# Impact of Facebook on Social Support and Emotional Wellbeing in Perinatal Women during Three Waves of the COVID-19 Pandemic in Mexico: A Descriptive Qualitative Study

**DOI:** 10.3390/ijerph20032472

**Published:** 2023-01-30

**Authors:** Ma. Asunción Lara, Laura Navarrete, Erica Medina, Pamela Patiño, Marcela Tiburcio

**Affiliations:** Dirección de Investigaciones Epidemiológicas y Psicosociales, Instituto Nacional de Psiquiatría Ramón de la Fuente Muñiz, Ciudad de México 14370, México

**Keywords:** perinatal, COVID-19, Facebook

## Abstract

The COVID-19 pandemic affected the mental health of pregnant and postpartum women in unique, unprecedented ways. Given the impossibility of delivering face-to-face care, digital platforms emerged as a first-line solution to provide emotional support. This qualitative study sought to examine the role that a closed Facebook group (CFG) played in providing social support for Mexican perinatal women and to explore the concerns they shared during the COVID-19 pandemic. A thematic analysis of all the posts in the CFG yielded nine main categories: (1) COVID-19 infections in participants and their families; (2) fear of infection; (3) infection prevention; (4) health services; (5) vaccines; (6) concerns about non-COVID-19-related health care; (7) effects of social isolation; (8) probable mental health cases; and (9) work outside the home. Participants faced stressful situations and demands that caused intense fear and worry. In addition to household tasks and perinatal care, they were responsible for adopting COVID-19 preventive measures and caring for infected family members. The main coping mechanism was their religious faith. The CFG was found to be a useful forum for supporting perinatal women, where they could share concerns, resolve doubts, and obtain information in a warm, compassionate, and empathetic atmosphere. Health providers would be advised to seek new social media to improve the quality of their services.

## 1. Introduction

The COVID-19 pandemic precipitated a global crisis, causing millions of infections and deaths. Data show that Mexico had a less effective response to the pandemic than comparable countries, with COVID-19 infection and death rates being among the world’s highest despite the fact that Mexico ranked near the bottom in testing [1]. Social distancing measures to control transmission led to the disruption of daily routines, financial difficulties, and negative consequences for people’s mental health [2,3].

Coping with stress and mental health during the pandemic has been an even greater challenge for specific vulnerable groups, such as perinatal women. Pregnancy and childbirth are life events requiring complex adaptation processes, which alter the emotional state of a significant number of mothers and can trigger anxiety and depressive symptoms or clinical disorders [4,5]. Mothers who gave birth during the pandemic showed greater acute stress than those who did so before the pandemic [6]. Shorey et al. [7] found that the pooled prevalence for antenatal anxiety, antenatal depression, and postnatal depression symptoms were 40%, 27%, and 17%, respectively. Nevertheless, the prevalence of mental health symptoms in mothers has fluctuated during the course of the pandemic as countries have moved to control the virus [6,8,9,10].

A narrative review showed that fear may have been one of the strongest psychosocial stressors associated with COVID-19, such as fear of contagion, the future, and losing one’s job in occupations where working from home was not an option [11]. Pregnancy-related stress significantly increased as a result of worries and fear about the pandemic [12]. Mothers worried about breastfeeding, neonatal care, and postpartum vaccination [13].

Psychosocial stressors may have been particularly acute in low-middle-income countries (LMIC), where many people were unable to isolate themselves as fully as required, probably for financial reasons, since most breadwinners (62%) were required to go to work at some time during the pandemic [14]. Negative feelings toward COVID-19 had an enormous impact on low-income pregnant women, tripling the prevalence of common mental disorders such as depression, generalized anxiety disorder, social anxiety, and post-traumatic stress disorder [15].

Studies of the effect of the pandemic on the mental health of Mexican women found higher scores for total stress and contextual stressors, such as financial problems and work difficulties, than in other population groups [16]. It was observed that significantly more women than men reported feeling worried and fearful [17], and that working women, such as physician mothers, who were mostly in partial lockdown, found that the number of hours they spent on paid employment, childcare, and food planning and preparation increased, while the amount of time they devoted to exercise and personal care decreased [18]. These findings showed that the pandemic had an unequal impact on women, with a greater toll on their mental health.

In regard to Mexican women’s mental health during COVID-19, 33.2% of pregnant mothers suffered stress and 17.5% depression [19]. A high level of stress during the pandemic was associated with higher levels of depression, whereas social support was associated with lower levels [20]. Other studies confirmed the high rates of mental disorders: 28.4% of mothers reported postpartum depression [21], while 46.1% experienced postpartum trait anxiety symptoms [22]. It is well documented that mothers’ mental health symptoms not only affect mothers but can also have short- and long-term negative effects on their infant children’s behavior, as well as their emotional and social wellbeing [23,24,25].

Potential factors affecting mental health due to the COVID-19 outbreak included social isolation, concerns about illness, distance from family and friends, loneliness, and changes in relationships [6,26,27]. There is evidence that women whose friends and relatives had COVID-19 were more likely to experience depression or anxiety symptoms [8]. Social support was one of the main factors associated with the mental health of perinatal women. Greater social support was a protective factor, particularly for pregnant women who viewed the impact of COVID-19 more negatively [28], while low levels of perceived social support increased depressive and anxiety symptoms [6]. The main issues related to stress were uncertainty surrounding perinatal care, the risk of exposure for both mother and baby, and inconsistent messages from information sources [29]. Changes in health care services also intensified pregnancy-specific stress, which indirectly affected perinatal mental health [30]. Lack of information on the effects of COVID-19 during pregnancy was associated with an increased risk of anxiety [6], and those whose regular checkups had been interrupted due to lockdown measures were more likely to experience depression or anxiety symptoms [8].

As in other countries, the pandemic had an enormous impact on health care provision in Mexico. Many hospitals only admitted patients with COVID-19, referring others elsewhere to limit the risk of transmission. The number of prenatal checkups was restricted and hospitals were repurposed for care delivery [31]. During the initial phase of the pandemic, many health centers and hospitals only admitted mothers; very few allowed fathers into delivery rooms, and visitors were forbidden [31]. Once mothers returned home, social distancing measures prevented visits from friends and relatives, which reduced social support and increased the risk of perinatal mental health problems.

Mental health services were already overstretched in many countries prior to the pandemic. In Mexico, only 6.4% of those with some form of mental illness receive the care they need [32] and mental health care is not considered a priority for perinatal women [33,34]. During the pandemic, pregnant and postpartum women therefore faced significant additional barriers to already scarce psychological treatment resources [35,36].

Against this background, the COVID-19 outbreak intensified global concern over strategies for promoting mental wellbeing and targeting the increased risk factors for mental illness [2]. Both mental health professionals and people suffering from depression, anxiety, and stress resorted to digital tools to provide or seek mental health care [37]. Many people had already had some experience with these tools in their everyday lives prior to the pandemic, and many were accessible on mobile phones, facilitating their widespread use [37]. These tools are helpful for accessing reliable data and information in real time and for receiving personalized recommendations.

Franz et al. [38] note that Facebook has been identified as one of the most influential online communication media. There is evidence that users regard social media platforms as reliable sources of information on pregnancy and COVID-19, and that some use Facebook to learn about COVID-19 from others [39]. Chatwin et al. [40] found that in a professionally mediated social media model of support, a high degree of trust and the ability to verify information and expertise had a direct impact on stress and anxiety levels in pregnant women. For specific mental health conditions, there are also numerous online communities and forums, such as Facebook groups, where patients can learn from the experience of others and receive peer support [41].

In short, pregnant and postpartum women in Mexico faced unique challenges during the COVID-19 outbreak, when the possibility of receiving mental health care for stressful situations and anxiety and depression symptoms was almost nonexistent. The main aim of this study was therefore to examine the role that a closed Facebook group (CFG) could play in providing social support and information to Mexican pregnant and postpartum women during the COVID-19 pandemic. Moreover, to our knowledge, only a handful of studies, none conducted in Latin America, have documented how perinatal women experienced pregnancy, childbirth, and caring for a new baby during the lockdown and reopening phases. A second aim was to explore the experiences, concerns, and emotional wellbeing during the pandemic that mothers shared in the CFG. The findings may provide guidelines for the use of social media during future crises to reduce stress by providing social support, and to prevent psychological symptoms from progressing to clinically diagnosed illness.

A qualitative analysis seemed to be the appropriate approach for these aims [42], as it would make it possible to reflect on what women were experiencing during COVID-19, as well as their spontaneous responses to the support and information offered via the CFG. Qualitative research conducted during previous pandemics yielded significant contributions to the understanding of the lived experience of these global events [43], since it enables one to analyze the context and behavior of people and their groups, from the point of view of those being studied [44]. Qualitative analysis is suggested when exploring a problem on which there is limited knowledge and seeking to identify the perceptions of health in specific groups to establish priorities [44]. In addition, this approach allows one to devote more time to the target group, in this case, the participants in the CFG and to focus on the characteristics of these interactions. This approach is suitable for conducting participatory research, where researchers also act as group facilitators, analyzing subjects’ concerns and difficulties based on their comments and responding to them [45]. Quantitative methodology is unsuitable for these aims.

## 2. Method

### 2.1. Study Design and Context

The qualitative study was based on an analysis of posts and comments in the CFG “Pregnancy and the Postpartum Period during the COVID-19” (https://www.facebook.com/saludemocionalperinatal; accessed on 10 December 2022). The profile was created two months after the COVID-19 lockdown in Mexico (March 2020) by a research group specializing in perinatal mental health, to provide information and emotional support to pregnant and postpartum women during the pandemic. Women could join the CFG if they said they were in the perinatal period and committed to the following guidelines: maintaining the confidentiality of subjects’ posts and comments, keeping the group a safe place where everyone felt free to express their emotions and experiences without being judged or rejected, and not making referrals to professionals in private practice or posting advertisements. The CFG facilitator, a clinical psychologist, commented on subjects’ posts a few times a day. Various hotlines for psychological support from public and private health/educational institutions that provided mental health care were offered. Due to the public nature of Facebook, questionnaires about personal data were not included; the main aim was simply to provide the service.

During the study period, 1707 women joined the CFG. Their ages ranged from 18 to 44, and 57% were Mexican, 34% were from Latin America, and 9% were from other countries. These demographic data were obtained from the statistics that subjects’ Facebooks provide in their open profiles, which were consulted respecting users’ privacy settings to limit access to their information. Considering that they have access to the Internet and their comments, we assumed that the self-selected sample was composed mainly of women with medium to medium–low socioeconomic status, who have their basic needs covered, as well as access to social health security, and in some cases to private medicine.

### 2.2. Data Analyses

All CFG posts (a total of 2680, including 349 from the facilitators and 2331 from the participants), from 2 May 2020 to 31 October 2021, were retrieved and pasted into a Word document and subsequently coded and categorized using thematic analysis [46]. Throughout this period, significant changes occurred during the COVID-19 pandemic, which was divided into five three-month stages: (1) COVID-19 lockdown (2 May 2020–31 July 2020); (2) reopening (2 August 2020–1 November 2020); (3) second COVID-19 wave (2 November 2020–8 February 2021); (4) vaccine available to vulnerable populations: health workers, those over 65, and those with chronic illnesses (9 February 2021–7 May 2021); and (5) vaccination of pregnant women and third wave of the pandemic (8 May 2021–31 October 2021). This division allowed the comparison of experiences and Facebook use across the stages of the pandemic.

The qualitative thematic analysis was used to analyze posts. It was undertaken in six steps. (1) All authors read the entire Word document with posts and comments to familiarize themselves with the content and participated in stage two. (2) Generation of preliminary categories and codes. (3) Search for the themes in each pandemic stage. (4) Cross-checking to obtain a consensus. Three authors were involved in stages three and four. (5) Second round of reading all posts and comments using the revised categories, definitions, and naming of topics. (6) Drafting of the report. All authors participated in stage five and made suggestions for the final draft. Repeated cross-checking and reviewing were central, as the initial coding was applied to the entire dataset, and solid support was required to ensure reliability [46].

The categories defined were: (1) COVID-19 infections in participants and their families; (2) fear of COVID-19 infection; (3) infection prevention; (4) health services; (5) COVID-19 vaccines; (6) concerns about non-COVID-19-related health care; (7) effects of social isolation; (8) probable mental health cases; and (9) work outside the home.

### 2.3. Ethical Considerations

According to the Council for International Organizations of Medical Sciences (CIOMS) [47], research may be exempt from being reviewed by an ethics committee when it is minimal risk and the data used for analysis are public and decoded. Data in this study met these characteristics in addition to protecting the identity of the subjects. The CIOMS also specifies that information from public websites may be used without requesting individual informed consent if researchers publish an announcement of their intention to conduct research, as was the case in this study. Posts were analyzed without any form of interpretation of personal data. The posts remained public both during and after the writing up of findings. The advantage of using Facebook data designed for public viewing is that it provides researchers with an enormous amount of data without the need to obtain informed consent (Franz et al. [45]).

## 3. Results

We will first describe the categories of situations subjects faced that created stress and anxiety during the COVID-19 pandemic, and subsequently analyze the group dynamics to find evidence of the support provided by the CFG.

### 3.1. COVID-19 Infections in Participantsparticipants and Their Families

As can be seen from Table 1, participants mentioned becoming ill with COVID-19 during pregnancy and in the postpartum period, as did their babies and other family members: “My whole family got COVID”. There were frequent posts like the following: “Hi moms; they just told me that I also tested positive for COVID. I am in my seventh month and very scared for my baby and my other children”. Another mother wrote: “I got COVID the first time before getting pregnant, but I hardly had any symptoms. This time, I was 22 weeks pregnant when I got symptoms, and by week 23 I was hospitalized”. The mothers were extremely anxious about testing positive: “Hi moms, I’m 15 weeks pregnant, and unfortunately yesterday I got the worst news in the world. My 10-month-old baby and I both got COVID and the emotions and everything are really hard, both for my baby and me”.

Infections close to the delivery date caused the greatest worry: “My baby will be born in two weeks. I don’t know how we got this [COVID-19] because my husband [only] went out for essentials. Sometimes I feel sad because this happened right at the end [of my pregnancy]”. The CFG was a space where women could share their experience of COVID-19: “I see that almost everyone here has got COVID right at the end of their pregnancy”. When mothers became infected, they worried about the consequences for the fetus and for nursing: “My baby is two months old. They gave me azithromycin… but they say I shouldn’t breastfeed while I’m taking it, but I’m afraid, first that my milk will stop and second that I will infect my baby”.

The women were responsible for looking after their partners, children, and other family members with COVID-19: “My father had symptoms, and I had no choice but to take care of him, and then I got sick… I didn’t think about the wellbeing of my baby or myself: we were both at the same risk of getting COVID, or even more so”.

### 3.2. Fear of COVID-19 Infection

The prevailing feeling was an enormous fear of getting COVID-19: “Honestly, I am panicking at the thought of getting COVID”. This was true especially as they neared their expected due dates. The women expressed their fears in different ways: “fear”, “sadness”, “worry”, “terror”, “uncertainty”, “anxiety”, and “stress”. They described these feelings as “intense” and “constant”, to the extent that they thought their emotional state had been greatly affected: “I cry all the time”, “I am overwhelmed by the situation”, “Bad luck could strike at any moment”, “I get stressed out”, “I get terrified”. The lack of knowledge about the new disease caused uncertainty: “We have been alright so far but I’m terrified of getting sick and having something happen right at the end [of my pregnancy]”. When there was a total lockdown except for essential tasks, the fear of infection was due to their partners’ having to go to work. When the reopening came, and employees were required to return to the workplace, this increased women’s fear that they themselves would be the source of infection: “I just think that hopefully this won’t happen to us, that when I have to go back to work I won’t infect my baby… [I have] no end of emotions”.

### 3.3. Infection Prevention

Women also assumed responsibility for implementing strict preventative measures: “I put a face shield [on my baby], I avoided getting close to anyone, I only did what was absolutely necessary and went straight home. I used hand sanitizer after any contact, and when I got home, I took a shower and bathed my baby”. In addition to these measures, some women took other precautions to take care of themselves, “to maintain their physical and mental health, such as eating healthily and getting enough rest”.

For some women, taking these measures seriously created problems with their families. “My husband sometimes thinks I overreact, because when he is called in to work, I insist that he take a shower when he comes home”. One husband looked for information to show that his wife was overreacting: “My husband started reading articles to prevent me from washing clothes so often”. Another kind of family problem was caused by mothers trying to protect their babies: “After the baby was born, I didn’t let anyone visit”.

### 3.4. Health Services

One area of concern was being able to find a hospital where women could have their babies delivered without the risk of COVID-19 infection: “Public clinics [are] where [women] are most at risk, since people go there for everything”. Women therefore looked for “a hospital with a maternity service free of COVID”. Opinions regarding public hospitals were mixed: “It’s good for us to have our babies there, since they are familiar with all their health problems, like gestational diabetes”. Most women looked for hospitals that did not treat people with COVID-19: “I found a hospital that had a maternity department free of COVID”. If they could afford to, they preferred to go to private hospitals, where their partners could accompany them. Those who had previously given birth noted that this time their experience was very different: “You [go into] the delivery room with a face shield and mask”. Family members were not allowed to visit. For women who had tested positive for the SARS-CoV-2 virus during labor, the situation was complicated: “They were admitted to the COVID department and had to stay in the hospital next to others who were infected, and watch those who were seriously ill dying”. Their babies were discharged immediately unless there were complications.

### 3.5. Vaccines

During the third and fourth stages, women began to share questions about getting vaccinated during pregnancy and nursing, although there were still no recommendations from either Mexican or international authorities. “Have any of you been vaccinated against COVID-19? I’m 25 weeks pregnant and want to get vaccinated”. “I’ve seen that expectant and nursing mothers should not get vaccinated, but I’m not a hundred per cent sure”. Specialist providers did not have much information either: “I have already been to two gynecologists and they told me not to get vaccinated”. The decision was left up to the women: “It’s my turn to get vaccinated tomorrow. I spoke to two obstetricians, and both told me it was my decision”. During the fifth stage, health authorities began vaccinating pregnant women, but it failed to include sufficient information. The exchanges on Facebook focused on which vaccine would be best for them as well as the side effects: “My baby is three months old today and is breastfeeding. I want to get vaccinated, but I’m afraid, because it’s time for his vaccinations at three months, so if I get vaccinated and they vaccinate him, could it be a problem for him?”

### 3.6. Concerns about Non-COVID-19-Related Health Problems

As described in Table 1, one concern at every stage of the pandemic has been the lack of vaccinations for children: “Hi moms, a question: Where did you get your baby’s first vaccinations, because the hospital where I gave birth doesn’t have them. Please help me”.

They also shared concerns about their babies’ health: “My baby was born with fetal tachycardia, and his heart could have stopped at any moment”. “My baby was born with a diaphragmatic hernia, and they have already operated on him, but it didn’t turn out well. They treated him again to give him dialysis, but it looks like his kidneys aren’t working”. They also shared problems such as bleeding during pregnancy.

### 3.7. Effects of Social Isolation

Being unable to leave the house and have contact with other people affected all the women, particularly when they had complications during pregnancy and were confined to bed: “Being shut in, the worry, and not being able to get out of bed is affecting my nerves”. For those who were pregnant, “not being able to physically buy things for my baby” caused them distress. The lack of social contact was also difficult: “Yes, I miss my mother’s hugs a lot, but we have talked a lot on video calls”. They also experienced a lack of support in household tasks once the baby was born: “We don’t have anyone to help us at home while I am recovering, so I rely on what my husband can do”. There were tensions in relationships with extended family members, since the women had to limit visits for fear that their babies would be infected: “For me, the most complicated thing has been keeping the grandparents and family from getting close to my baby”. “Sometimes relatives get offended, but you have to calmly try to make them understand and ensure that they not [visit], because you are afraid”. Their fears concerning social isolation also centered on the effects of the latter on the baby: “I don’t know how I am going to raise my baby in isolation, limiting contact with her surroundings, since that is also part of her psychosocial development”.

### 3.8. Possible Clinical Cases of Mental Illness

The emotional distress women experienced is clear from their testimonials. Throughout the five stages, there were only four cases of mental disorders diagnosed by specialists whom they had consulted for various problems: (1) borderline personality disorder: “The past few weeks have been really difficult, everything makes me cry. I don’t have any desire to get out of bed, I don’t even want my partner close to me, because I don’t know how to explain what is happening to me while I’m crying”; (2) prenatal depression: “I am pregnant now and getting psychological treatment, for prenatal depression, apparently”; (3) postpartum depression: “After my daughter was born, I started to have chest pains and cried a lot. I went to see my gynecologist and he told me it was postpartum neurosis. Well, you do what you can, and I’m gradually dealing with my attacks”; and (4) bipolar disorder: “I was severely depressed, since I have bipolar disorder. For that reason, they watched me very carefully while I was pregnant, and I have to take medication for the rest of my life. There are thoughts you can’t control with depression, but when you take the medication. it changes your life for the better”.

There were also two cases which, based on the symptoms reported, were considered possible clinical cases. One was possible prenatal depression. Facilitator: “If you have anxiety and depression, it is important to see your psychiatrist. Between your gynecologist and your psychiatrist, you will be able to find the appropriate treatment, and then both of you [you and your baby] will be safe”. One mother was grieving the loss of her baby during pregnancy: “I don’t even know how to express my feelings. My husband and I planned to have a second baby, and at the end of the year we found out that we had succeeded… They told me two things, one that made me very happy, that I was going to have twins, but the other was that it was high risk… The babies died, and since then I have been very sad, angry, stressed, and I blame myself for not having done more… They recommended psychotherapy, but my appointment was canceled because of the pandemic”.

### 3.9. Work outside the Home

For some women, working from home caused “moments of anxiety and stress because of the amount of work involved”. Those who began to return to work, even if they adopted all the self-care measures, “lived in a state of paranoia and always felt they had a sore throat”. Their families criticized them for putting their own and their babies’ health at risk by going to work. Those who did so were asked “whether they didn’t love the baby and themselves, and whether they would infect their families by going to work”.

The greatest anxiety during all the stages was the lack of clarity regarding maternity leave. During the first two trimesters of pregnancy, the possibility of working at home depended on the “flexibility of each employer, regardless of whether there had been COVID cases at the business”. The degree of stress over the return to work was related to the type of work: “My job is to help people, have contact with them, take their pictures without a mask”. Women were also unaware whether they could stay at home while they were breastfeeding to avoid putting their babies at risk. The attitude of the staff at certain health centers was insensitive: “I saw a medical director, and she rudely told me that pregnant women had to go to work as usual and that they were not a vulnerable group”. Some women had to return to work once they had completed the vaccination scheme: “When I’ve had both doses, I’ll have to go back to that life… I’m very scared, I’m stressed out, anxious, I’m always on edge”.

### 3.10. Group Dynamics

As these descriptions show, perinatal women experienced extremely stressful situations during this period. The following sections describe how the facilitator and the participants responded to these experiences.

(a) The Facilitator. At every stage, the facilitator reviewed posts and provided any comments she considered necessary a few times a day. During the first stage, she encouraged women to participate (“We would like to listen to you”) and reiterated the purpose of the CFG (“This space is for you, to accompany you throughout the pandemic”). She provided suggestions for managing emotions, such as “breathing and relaxation exercises, mindfulness, meditation, keeping a journal or talking to a trusted person, as well as keeping active and/or exercising”. She encouraged them to “keep in touch with your loved ones by phone, texting, or social media”, as well as by “creating virtual support networks with other mothers and with institutions”. She took care to thank the women for their participation.

The facilitator expressed empathy in all her interventions with the situations women were experiencing. “We are sorry you are experiencing this situation”. “Right now, because of the pandemic and isolation, pregnant and postpartum women have had to experience many losses, absences, shortages, changes, and adjustments that have caused stress, anxiety, fear, uncertainty, frustration, pain, and a lack of motivation, among other emotions”. She reiterated that she was there to support them. The participants thanked her for being in the CFG: “Thank you for accepting me into the group. Thank you very much”. The facilitator was extremely active during the first three stages, but the women gradually began to participate more, and in the last two stages “they fully appropriated the CFG”.

(b) The Participants. At first, there was limited participation by members of the group, but they encouraged one another. Over the five stages of the pandemic, they showed trust in one another, sharing important problems: “I had a difficult pregnancy”, “This situation we are experiencing is very confusing and complicated”, “I am very worried because my husband has COVID symptoms”. They supported one another at all times: “Relax, everything is going to be alright. God will help us get through it”. The majority recommended actions such as thinking positively, not focusing on negative emotions, entrusting themselves to God, and prayer: “Don’t lose faith. Leave everything in the hands of God”, “I trusted God and the doctors who reassured me”.

The women were very empathetic in their participation: “The same thing happened to me”. They addressed each other affectionately: “Hi beautiful”, “Hi mommies”, “Darling, I’m just the same”, “The same thing just happened to me”, “I think you’ll understand me”, “Yes, I understand you”, “Don’t blame yourself, it’s normal”, “Relax, I had COVID when I was pregnant and my baby is fine”. They shared information that had been useful to them: “Don’t self-medicate”. “Talk to your doctor”, “Let’s write a sentence every day that motivates us”, “Take care of yourself; I really didn’t think I was going to survive”, “Have the best possible hygiene when you nurse your baby”, “Don’t stop nursing him”. They also shared information to encourage those who were afraid of being vaccinated. They found reading each other’s comments very helpful: “reading what you shared somehow makes me feel more secure in this experience (of the pandemic), which is totally new for many people”, “it is very comforting to know that I am not the only one”, “what everyone says is very reassuring”.

The women thanked each another for their advice and for being there: “Thank you to everyone who has commented”, “Thank you for sharing your experience”, “Thanks to each and every one of you. Believe me, your words are comforting. It is a very difficult situation I wouldn’t wish on anyone, your words are encouraging”, “Thank you from the bottom of my heart”.

Apart from these expressions of gratitude, they also felt understood and accompanied. “When I joined this group, I was like you, I had many fears. By reading [the posts] I was able to clarify doubts I had. It encourages me that we are here to support and help each other”. They shared their satisfaction with the experience. “They recommended this group so I could receive support and I really like it, because I no longer feel alone”.

## 4. Discussion

The aim of this study was to examine the role that Facebook played in providing information and social support for Mexican pregnant and postpartum women and to explore the experiences and challenges they shared in the CFG during the COVID-19 pandemic.

Pregnancy and childbirth are life events that always require a complex adaptation process, which affects the emotional state of a significant number of mothers. From the comments and posts in the CFG, it was clear that for Mexican perinatal women, coping with COVID-19 was extremely challenging and demanding. Our findings are in line with other researchers’ observations about the COVID-19 pandemic being difficult for women globally, and particularly stressful for perinatal women, as they faced uniquely demanding situations during this period [48,49].

Becoming infected with COVID-19 elicited intense emotional reactions. The most stressful period during the pandemic for the women in our sample was the outbreak, with the sudden lockdown, changes in health systems, and uncertainty about the new disease and its effects on fetuses and newborn babies, which is similar to what has been found in other countries [24]. Hendrix et al. [50] note that the first months of COVID-19 were possibly the most universal disruption to perinatal care in recent history, and our results support this observation.

Several posts in the CFG related to mothers being infected with COVID-19, and the fear, stress, and worry they experienced. They were concerned about the possible effects of the virus on their fetuses, and how it might alter their childbirth and breastfeeding plans. Becoming infected close to their expected due date was even more distressing, as it meant they would be separated from their newborn babies. Intense, constant, acute fear was one of the strongest feelings experienced among the participants, which they also described as worry, anguish, and stress. The fear and stress extended to all situations, not only to possible infection, but also to the future, having to go out to work or take the baby to the doctor, and to the effects of COVID-19 on the baby’s health. There is ample evidence documenting the deep fears experienced by pregnant and postpartum women in most countries as a result of the emergence of this novel virus [13,24,51].

A study comparing pregnant and postpartum women living on various continents found that North Americans and Latin Americans (including Mexicans) were more likely to have increased levels of worry [49]. In the case of Mexico, this greater level of concern could be related to the country’s poor management of the pandemic, as compared with other countries [3], and to other local factors, such as the shortage of childhood vaccines, which had begun before the pandemic and had not yet been resolved.

Women were responsible for implementing COVID-19 prevention measures and caring for infected partners, babies, and other family members, risking their own health. When isolation removed their support systems, such as family members and friends, they lacked help with household chores even after their babies had been born. What little help they received with household chores and caring for the baby was from a partner “who did what he could when he returned from work”, at a time when partners were “being subjected to excessive work and inflexible working hours” [52]. Many women were also employed, either working from home or going to work, when their maternity leave expired. Taken together, this meant that perinatal women’s usual multiple roles increased, further widening the gender equity gap [53].

Women’s traditional role of prioritizing the needs of others over their own [54] appears to have been exacerbated during the pandemic, which in turn impacted their emotional health, as has already been observed in this population [19,20,22,52]. Data on Mexican women in general during the COVID-19 pandemic also show that women were more stressed than men [16,17], and that the unequal division of responsibilities between men and women was exacerbated [18].

Social isolation meant that women were denied affection from their loved ones, including their mothers, and they also feared for the psychosocial development of their infants, who were deprived of contact with other people. They assumed that this lack of socialization could be harmful in the long term to their babies’ mental health, as has been observed in other studies [52,55]. Nevertheless, they restricted visits for fear that their babies would be infected, which strained family relationships. This was particularly true where mothers took strict measures to prevent their babies from getting infected, which their relatives considered extreme.

As in most countries [49], prenatal care, as well as delivery procedures, was relocated to facilities separate from those that treated patients with COVID-19. These readjustments increased the barriers to prenatal and postpartum care. Even though women expressed fears that they or their babies could become infected in health care facilities, they did not express complaints about the delivery care they received. Their main complaint was the uncertainty about which hospital would be assigned to them to provide delivery care, the effects of COVID-19 on their health and that of the fetus and baby, and the question of whether they should be vaccinated during pregnancy and lactation. From the outset, the Ministry of Health’s Linea Maternal hotline was insufficient, and the information it provided was often contradictory. A similar situation was faced by women in other countries [55]. For instance, during the pandemic in Canada, it was difficult for mothers to access information and support. Many of them reported they lacked information on how to take care of their babies, including information on safe breastfeeding and public health regulations.

Working women who were required to return to the workplace were extremely worried about getting COVID-19, even if they followed all the recommended self-care measures. They also experienced great distress about the lack of clarity regarding maternity leave. There were no special considerations for pregnant and postpartum women in the granting of maternity leave, which raised doubts about the adequacy of the measures to protect working women’s health and mental wellbeing during the COVID-19 pandemic.

Four women in the CFG were clinically diagnosed by their mental health providers, and two others presented with clinical mental disorder symptoms during this period. Participating in the group gave them the opportunity to be heard and receive support from the facilitator and the other participants, which may have delayed the progress of their disorder. There is evidence that being part of a face-to-face group gives perinatal women a sense of belonging and of not being the only ones experiencing emotional symptoms [56]. Realizing that one is not the only one experiencing this type of emotions and feelings is regarded as an important therapeutic factor in group therapy [56].

One important consideration in this context is the role of the women’s faith and religious convictions as a mechanism for coping with their fears, illness, anxiety, and uncertainties. This is consistent with the observation that Mexican culture has a strong religious tradition. Some studies claim that strengthening spiritual behavior during the COVID-19 pandemic produced dramatic changes in people’s lives [43].

Regarding the role of the CFG in providing social support and information, the data show that it was effective in terms of time, costs, and immediate response to perinatal women during the COVID-19 pandemic. Some elements that contributed to the positive result were the facilitator’s solid background and experience in perinatal mental health, and her monitoring of the group to ensure adherence to the guidelines, such as showing respect and avoiding lecturing others, as well as modeling an empathetic, respectful, and caring attitude. The women’s response in making the group their own, eagerly exchanging suggestions, and showing empathy and understanding toward the problems of others also underlined the important ways in which a CFG helped support perinatal women. Subjects’ many expressions of gratitude confirm that they felt understood and accompanied. This type of response has been observed by women in other Facebook groups. In one, for example, a woman remarked: “[It´s] just been great to have support from the other mums all going through the same thing…”; [40]. In short, our findings are in keeping with those of other studies that social media sites such as Facebook can serve as an effective means of providing social support and information [41,56,57].

In a broader context, the COVID-19 pandemic produced a host of negative consequences, but at the same time showed the potential of providing mental health care services, supplementing face-to-face support with the use of new technologies. The closure of mental health care services affected the delivery of emotional support and treatment to perinatal women, which was deficient even before the pandemic [34]. This meant that the effects of COVID-19 on Mexican perinatal women’s mental health could not be addressed through hotlines, which proved insufficient from the outset. Conversely, social media tools demonstrated their enormous potential for delivering social support and information in a crisis, when all types of resources are limited.

## 5. Conclusions

This qualitative descriptive study showed that Facebook was an efficient means of providing social support and information in the context of the pandemic to perinatal women. It was considered effective due to the small number of resources required to set it up and to the easiness to socializing our Facebook among the target population due to the enormous acceptance Facebook has for seeking information and help. It was also considered effective due to the numerous spontaneous expressions of gratitude for receiving advice and information from the participants. Our findings also revealed what Mexican pregnant and postpartum women experienced during COVID-19. Their comments reflect the numerous stressful situations they faced due to the changes in perinatal care; the confinement measures to limit the expansion of infections; the fear of themselves, their babies, and their families becoming infected; and the uncertainty about maternity leave and having to return to work, among others.

One of the strengths of the study is that it examines the experiences and concerns of expectant and postpartum mothers from their perspective, at the time when they occurred, reducing memory bias. At the same time, it reveals the potential of Facebook to provide support and information for perinatal women to reduce the perinatal mental health care gap. To the best of our knowledge, this is the first published study on Facebook in Mexico used with this population for this purpose.

### 5.1. Limitations

Limitations of this study include the fact that the data are drawn from a self-selected sample, composed mainly of women with medium to medium–low socioeconomic status, who have their basic needs covered, as well as access to social health security, and in some cases to private medicine. Our findings do not therefore represent the experiences nor the usefulness of digital devices for more disadvantaged women with limited access to the Internet and electronic devices, who lack more efficient health services, and were also more severely affected by the pandemic. This may explain why none of them reported domestic violence or difficulties with their partners, which was a common finding in Mexico during this period.

### 5.2. Implications for Perinatal Mental Public Health

Social networks such as Facebook clearly have the potential to serve as a source of social support to vulnerable populations. Social media tools that provide mental health services are here to stay. It remains for Mexican institutions responsible for the health and mental health of the population to incorporate them. There is also a need to develop specific protocols to reach groups that have fallen between the cracks.

## Figures and Tables

**Table 1 ijerph-20-02472-t001:** Major themes by COVID-19 pandemic stage in a Facebook group for Perinatal women.

	Stage 1 2 May–31 July 2020	Stage 2 2 August–1 November 2020	Stage 3 2 November 2020–8 February 2021	Stage 4 9 February–7 May 2021	Stage 5 8 May–3 October 2021
Category	First wave of infections, 23 March: pandemic containment measures.	Infections decrease, partial opening of essential activities.	Second wave of infections.	Vaccination of older adults begins.	Third wave of infections, vaccinations of pregnant women.
1. COVID-19 infections in participants and their families.	- Only one pregnant woman (and her partner, parents, and in-laws) report infection.	- Infections increase among women, partners, and families. Insufficient data regarding prenatal effects of COVID-19.	- High infection rate among women. - Some hospitalizations of women and their families. - Seeking information on after effects of COVID-19.	- Infections decrease in the general population, yet remain constant in women. - Hospitalizations	- New infections in women, including second infections and even in vaccinated women. - Infections of babies.
2. Fear of COVID-19 infection.	- Acute fear of infection because of the possible effects on pregnancy, childbirth, and the newborn baby. - Uncertainty about the future. - Fear of newborn baby becoming infected in hospital.	- Anxiety regarding infection on return to work. - Stress over possible consequences of COVID-19 medication during pregnancy and nursing. - Sadness over infections. - Difficulties with family members who do not take COVID-19 precautions.	- Terrified of infection. - Fear of consequences of COVID-19 medication during pregnancy and nursing. Fear of returning to work and using public transportation. - Worry about their other children. - Fear of “losing the baby” through infection. - Greater anxiety in the third trimester of pregnancy.	- Worry about reinfection and use of medication. - Anxiety over suspicion of infection in themselves and/or their partners. - Worry over possible effects of the virus on the fetus. - Despite the decrease in cases, fear of the severity of symptoms and the effects of medication during pregnancy.	- High level of fear and emotional unease about the possibility of infection. - Fear of possible vaccine side- effects.
3. Infection prevention	- Women assume responsibility for adopting preventive measures involving hygiene and isolation. - Problems with those who do not follow preventive measures. - Skip medical appointments to prevent contagion.	- Care for sick family members, risking their own health and that of their baby. - Measures taken to avoid infection: isolate themselves as much as possible, cleaning with bleach and alcohol, wearing masks, constant handwashing.	- Consciously maintain preventive measures. - Share the measures they adopt when they go to work. - Isolate themselves from infected persons. - High level of stress due to adopting preventive measures.	- Recommend healthy diet and social isolation. - Disinfection of home after COVID-19 infection.	- Focus on the issue of vaccines and their side effects on them and their pregnancies.
4. Health services	- Suspend prenatal care appointments. - Priority given to women in the third trimester. - Changes in hospital previously assigned for childbirth. - Some women switch to private hospitals. - Women stop going to postnatal care appointments. - High level of concern about giving birth in COVID-19 hospitals.	- Note differences with respect to previous births (use of face shields and masks). Differences in care in COVID-19 hospitals vs. non-COVID-19 hospitals. - Priority given to appointments in the third trimester. - Difficulty finding hospitals that do not treat COVID-19.	- Search for hospitals offering childbirth procedures for women with COVID-19. - Changes in handling childbirth when there is a COVID-19 diagnosis. - Isolation of newborns from mothers with COVID-19. - Medical appointments without partners. Longer intervals between prenatal care appointments.	- Longer intervals between prenatal care appointments than before the pandemic. - Search for non-COVID-19 hospitals for delivery. - Partners now allowed to enter private hospitals.	- Fear of going to hospital because of high infection rates.
5. COVID-19 vaccines			- Beginning of discussion about vaccines, although not about whether they are authorized for pregnant women. - Exchange of information about what to do about vaccination.	- Vaccination begins for health care personnel, older adults, and teachers. - Contradictory information among health professionals regarding vaccination of pregnant women. - Decision left to mothers.	- Vaccination of pregnant women approved by the WHO. - Relevant information about vaccination of pregnant women not provided. - Despite their fear, pregnant women want to be vaccinated. - Sharing of information about vaccines and side effects.
6. Concerns about non-COVID-19-related health problems	- Concern about shortage of infant vaccines (BCG). - Pregnancy-specific fears. - Threat of premature birth.	- Concern about shortage of infant vaccines. - Difficulties with nursing in crowded homes.	- Concern about shortage of infant vaccines. - High-risk pregnancies. - Fetuses with health complications.	- Health problems of newborns. - Health problems in pregnancy and in nursing.	- Share information on effects of vaccines.
7. Effects of social isolation	- Unable to prepare for baby’s arrival. - Desire for contact with significant persons so they can see the baby. - Worry about lack of socialization for the baby. - Lack of support. - Miss direct personal contact.	- Although it is now possible to go out, many remain in isolation to protect the baby’s health.	- Some remain in isolation to protect the baby’s health.		
8. Probable mental health clinical cases	- Probable case of post-partum depression.	- -Borderline personality disorder. - Grief at death of fetus.			- Prenatal depression. - Postpartum depression. - Bipolar disorder. - Possible prenatal depression.
9. Work outside the home	- Fear of going to work, becoming infected with COVID-19, and infecting the baby. - Unclear whether they are considered a vulnerable population with the right to longer maternity leave. - Anxiety about application process for maternity leave.	- Greater fear of infection with reopening.	- Maternity leave only granted under existing law with no modifications due to COVID-19. - Rejection by family for continuing to work, putting mother’s and baby’s health at risk. - Unable to stop working because source of family economic support.	- Unclear whether they are considered an at-risk population. - Some companies allow working from home, without the need for the authorization of maternity leave.	- Continued questions about maternity leave. - Still not considered an at-risk population. - Some mothers obtain permission to work from home until they complete the vaccination scheme. - Fear of returning to work.

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
