# Peer review of "Impact of Facebook on Social Support and Emotional Wellbeing in Perinatal Women during Three Waves of the COVID-19 Pandemic in Mexico: A Descriptive Qualitative Study"

_ijerph, 2023, doi:10.3390/ijerph20032472_

Round 1

Reviewer 1 Report

In this manuscript, the researchers describe the Impacts of facebook health grouping providing emotional and social support to perinatal women. The availability of social media and its contents have evolved and are a crucial point of impact in health care. I have the following questions

1. How did the authors verify that the participants were perinatal women?
2. Kindly mention in detail how many participants were involved? average (minimum, maximum) duration of participation per participant? mention age? parity? education? vocation? social support? 
3. Was there any feedback questionnaire or assessment scoring to identify if the above method benefited the participants?
4. Was a satisfaction level assessment done after the intervention
5. Authors have shown that patients were able to share their concerns but I am not sure how they are conveying that these concerns were graded for significance and addressed. 
6. Telephonic communication has been significant in the present scenario for follow-up during the epidemic. Mention the hindrances in Facebook-based follow-up. Refer to and cite PMID: 27306362

Author Response

Dear Referee:

We appreciate your helpful suggestion. We have addressed them as fully as possible.

Reviewer 2 Report

The abstract must include clear methodology, results and conclusions. The current abstract is generic and broad.

Line 166, remove the initials of authors.

Overall, the manuscript is not scientific. The variables and study design are so weak and cannot be interpreted right. 

Author Response

Dear Referee:

We appreciate your thoughtful comments as they gave us the opportunity to add needed information to further explain the methodological approach used.

Extensive editing of English language and style required:

Author Response

Dear Referee:

Thank you very much for your positive comments, they were very useful to improve our manuscript.

Round 2

Reviewer 2 Report

NA